# Camelid Single-Domain Antibodies: Promises and Challenges as Lifesaving Treatments [note 1]

**DOI:** 10.3390/ijms23095009

**Published:** 2022-04-30

**Authors:** Mehdi Arbabi-Ghahroudi

**Affiliations:** 1Human Health Therapeutics Research Centre, National Research Council Canada, 100 Sussex Drive, Ottawa, ON K1A 0R6, Canada; mehdi.arbabi@nrc-cnrc.gc.ca; 2Department of Biochemistry, Microbiology and Immunology, University of Ottawa, Ottawa, ON K1H 8M5, Canada; 3Department of Biology, Carleton University, 1125 Colonel By Drive, Ottawa, ON K1S 5B6, Canada

**Keywords:** camelid heavy-chain antibody, VHH, single-domain antibody, nanobody, caplacizumab, biotherapeutics, camelid mice, bispecific VHH

## Abstract

Since the discovery of camelid heavy-chain antibodies in 1993, there has been tremendous excitement for these antibody domains (VHHs/sdAbs/nanobodies) as research tools, diagnostics, and therapeutics. Commercially, several patents were granted to pioneering research groups in Belgium and the Netherlands between 1996–2001. Ablynx was established in 2001 with the aim of exploring the therapeutic applications and development of nanobody drugs. Extensive efforts over two decades at Ablynx led to the first approved nanobody drug, caplacizumab (Cablivi) by the EMA and FDA (2018–2019) for the treatment of rare blood clotting disorders in adults with acquired thrombotic thrombocytopenic purpura (TPP). The relatively long development time between camelid sdAb discovery and their entry into the market reflects the novelty of the approach, together with intellectual property restrictions and freedom-to-operate issues. The approval of the first sdAb drug, together with the expiration of key patents, may open a new horizon for the emergence of camelid sdAbs as mainstream biotherapeutics in the years to come. It remains to be seen if nanobody-based drugs will be cheaper than traditional antibodies. In this review, I provide critical perspectives on camelid sdAbs and present the promises and challenges to their widespread adoption as diagnostic and therapeutic agents.

## 1. Introduction

Immunoglobulins (Igs) or antibodies are the key elements of the adaptive immune system. These Y-shaped bifunctional glycoproteins can specifically recognize non-self-antigens, thus forming a molecular communication bridge with other elements of the immune system to neutralize and eliminate foreign pathogens. In response to signals from the innate immune system, Igs are developed through a precise molecular mechanism and are first displayed on B lymphocytes’ membrane and then secreted into bodily fluids. The adaptive immune response paradigm has been reasonably conserved in mammals, though antibody structure and isotype variations do exist across species [1]. In human serum, for instance, immunoglobulin G (IgG) is the most common isotype and IgG-based monoclonal antibodies (mAbs) have been utilized extensively in research and medicine over the past several decades, with a notable 100 IgG mAbs approved as drugs as of December 2021 [2,3]. IgG isotypes are further divided into four subclasses (IgG1-4) based on their structure and function with IgG1 the most abundant subclass in serum. The IgG molecules are composed of two identical heavy chains and two identical light chains forming a tetrameric structure. The heavy chains and light chains are composed of four domains (VH, CH1, CH2, and CH3) and two domains (VL and CL), respectively. The antigen-binding properties of antibodies are predominantly concentrated in short segments within the variable domains (VH and VL) which exhibit a high degree of variability. These are called hypervariable regions/loops or complementarity-determining regions (CDRs) in view of their direct involvement in antigen binding (Figure 1) [4,5,6].

These domains adopt a conserved Ig fold structure which consists of two layers of anti-parallel β-plated sheets closely packed with hydrophobic amino acid side-chains, resulting in a β-barrel or sandwich-like structure [7,8]. The C-terminal region of the IgG (known also as crystallizable fragment or Fc) including CH2 and CH3 domains is well conserved and engages in various effector functions. The fragment for antigen binding (Fab arm) includes the two variable domains (VH and VL) followed by two constant domains (CH1 and CL). The Fab arm is also spaced from the Fc stem by a hinge region (with various lengths among IgG isotypes) which has an essential role in allowing the Fab to rotate as much as 158° and the Fab-Fab and Fab-Fc to acquire angles that range from 115–172° and 66–123°, respectively [9,10].

The conventional view of antibodies changed dramatically in the past few decades when an in-depth analysis of the structure, genetics, and functional binding of Ig variants in several mammalian species was performed. Of particular interest were species including old- and new-world camelids: *Camelus dromedarius* (Arabian camel), *C. ferus* (wild Bactrian camel), *C. bactrianus* (Bactrian camel), *Lama glama* (llama), *L. guanaco* (guanaco), *Vicugna* (vicugna), and *V. pacos* (alpaca). In the 1990s, a research team at the Vrije Universiteit Brussel (VUB), led by Professor Raymond Hamers, analyzed the total and fractionated sera obtained from healthy dromedaries in Morocco and observed smaller IgG subclasses that seemed to lack the light chains. The first set of data published in 1993 showed that dromedaries’ serum, besides the conventional IgG (thereafter called IgG1; MW~150 kDa), contains two smaller immunoglobulin fractions (thereafter called IgG2 and IgG3; MW ~90 kDa) lacking the light chains and the first heavy-chain constant domain (named thereafter: heavy-chain-only antibodies or HCAbs) and contributed up to 75% of total serum IgG (Figure 1). Subsequent studies on other camelid species (llama and alpaca) demonstrated the existence of HCAbs although at lower concentrations (25–50%). Most importantly, in this report, the antigen-binding activity of HCAbs was demonstrated from a dromedary exposed to *Trypanosoma evansi* by radio-immunoprecipitation and blotting experiments [11,12,13].

Initial antibody gene cloning studies revealed the unique sequences of the variable domain regions of HCAbs. In 1994, we decided to use the term “VHH” for the variable domain of camelid variable domains to distinguish it from the conventional VH domains with important hydrophilic amino acid substitutions at key contacting residues with the former VL domain, namely, V37F/Y, G44E, L45R, and W47G (Kabat numbering) [14,15]. Additional cDNA sequencing from camelid lymphocytes helped to generate universal primers for the cloning of the heavy-chain repertoire [16,17]. Subsequently, an immunized dromedary repertoire was used to build a phage display VHH library to examine the feasibility of isolating camelid VHHs by phage panning against two model antigens, lysozyme and tetanus toxoid. After four rounds of panning and phage ELISA screening, VHHs specific to lysozyme and tetanus toxoid with nanomolar affinities were isolated and characterized. VHH solubility was demonstrated by size exclusion chromatography, showing that VHHs were monomers without a propensity to aggregate. The circular dichroism spectrum (between 197–215 nm) of the concentrated VHHs showed a β-pleated sheet fold as expected for a well-structured Ig domain [18] (Figure 1c).

Following the discovery of HCAbs in camelid species, the presence of a similar class of heavy-chain immunoglobulin, named IgNAR, was found in the adaptive immune system of a distantly related species of nurse shark and reported by Greenberg and co-workers in 1995 [19]. These heavy-chain antibodies have one variable domain, called the variable domain of new antigen receptor (vNAR), followed by five constant domains and exist in all elasmobranchs [20]. Immunological and structural studies on vNAR fragments showed that they have similar biophysico-chemical properties to camelid VHHs, including small size, high affinities and specificities, high thermal stability, and low production cost. These antibody fragments are considered promising antigen-binding domains which will have important medical and biotechnological applications in years to come. Discussion on these heavy-chain antibodies and vNAR fragments is out of the scope of this review and interested readers are referred to several excellent reviews published recently [21,22].

## 2. The Structure of Camelid VHHs

Resolving the apo crystal structure of the VHH and a VHH:lysozyme complex revealed rather unique features of an anti-lysozyme VHH where the N-terminal segment of CDR3 penetrated into the catalytic site of lysozyme and the CDR3 C-terminal end folded over the former VL interface, shielding some of the remaining hydrophobic area (e.g., Phe 42) in the VHH. This type of antibody:antigen interaction required a major shift from the previously established canonical structures for human CDR1 and had hardly ever been observed before with conventional antibodies [23,24,25,26,27]. Over the next two decades, additional structural studies showed that there were a number of alternative VHH–antigen interactions beyond the recognition of enzyme active sites and other cavities on the protein surface. These include the recognition of flat surfaces on proteins and even the ability of a VHH to form a convex paratope to accommodate binding to small molecules and haptens [25,28,29].

To draw a comprehensive picture of VHH:antigen interactions, a recent structural study on 105 VHH-antigen crystal structures with resolution <3 Å revealed that VHHs tend to recognize more rigid, concave, conserved, and structured epitopes enriched with a combination of aromatic, polar, charged, and hydrophobic amino acids. This study also found that CDR3 was involved in about 50% of all interactions whereas CDR2 and CDR1 contribute to 20% and 13% of all interactions, respectively. Most importantly, the share of non-CDR residues reached 16% of all interactions and was located in four distinct regions on the VHH scaffold (Figure 2), namely, the N-terminal region of the first β-strand (A), FR2 region, the C” and D loop, and the DE loop (also known as CDR4). To conclude, it appears that the VHH paratope makes use of every accessible residue in the “VHH:antigen interaction cloud” to strengthen binding affinities [30]. In comparison, a separate study of 227 structures of conventional antibody-antigen complexes found that 80% of interactions are limited to 4–13 residues. Furthermore, 3 out of 30 highlighted positions that have major contributions to binding affinity are located in framework regions and the remaining 27 are located within CDR regions (defined by Kabat [31,32]), suggesting that conventional antibodies rely more heavily on binding via amino acids in the CDR regions than HCAbs do. Taken together, these findings suggest that VHH:antigen interactions deviate from those typically seen for conventional antibody–antigen interactions and are more similar to protein–protein interactions [30].

We believe that evolutionary removal of the VL domain established the VHH as a novel antigen-binding domain where the conventional role of CDRs has been somewhat modified and the binding interactions are extended to non-CDR regions. At the same time, a number of evolutionary mechanisms such as extending the length and complexity of amino acids in both CDR1 and CDR3 loops, participation of non-CDR regions including residues in FR2 and “CDR4” (residues 76–80; Kabat numbering) in antigen binding and also shaping the CDR3 loop, and increasing the rate of somatic hypermutation which is concentrated, but not limited, to CDR regions that have been applied to enlarge the VHH repertoire diversity in the absence of the VL domain [27]. Consequently, each and every amino acid position in the VHH scaffold has increasing importance, both for structural integrity and antigen-binding, compared to the equivalent residues in a classical VH which otherwise share these characteristics with the paired VL domain.

## 3. The Genetic Origin of Camelid HCAbs

Genomic studies, cDNA sequencing, and next-generation sequencing of camelid heavy-chain antibody repertoires have shown that HCAbs possess unique functions in the camelid adaptive immune system and emerged through common ancestry with a precise molecular evolutionary mechanism. The highlights of these findings are: (a) the first constant domain (CH1) is spliced out in HCAbs, leading to direct attachment of recombined VHH exon to the hinge region with various lengths (0–27 aa); (b) specific amino acid substitutions are imprinted in the camelid genome in FR2, primarily on the side of the VH that contacts the VL in classical antibodies, including at Kabat positions: V37F/Y, G44E, L45R, and W47G (V42F, G49E, L50R, W52G; IMGT positions), as well as in some other positions in FR1 and FR3 (such as L11S, P14A, and A83P) which would contact the former CH1 domain in conventional IgGs; and (c) the length of CDR regions (loops) is expanded in HCAbs and deviates from the canonical loop structures established for human and mouse Igs [27,33]. For instance, VHH CDR3 shows a broader distribution of lengths (3–28 aa) relative to those reported for the VH domains of conventional mAbs in other species such as human, mouse, and rabbit (2–19 aa) [27,30,34,35,36,37,38].

Immunogenetic studies have shown that camelid HCAbs have preserved unique evolutionary mechanisms to diversify the VHH paratope repertoire, including: (a) a relatively large pool of VHH minigene segments that recombine with two smaller DH and JH minigenes along with a higher rate of insertions/deletions at the V-D-H junctions; (b) higher incidence of recombination signal sequence (RSS) in FR3 region (residue 76-78) underpinning the phenomenon of gene replacement which involve recombination activating-gene (RAG) protein; (c) higher rates of somatic hypermutation in CDRs, including a novel hot spot hypermutation codon (TAY) in CDR1, and further extending to non-CDR regions, in particular to FR3; (d) acquirement of additional cysteine residues within the FR2 and CDR region; and (e) participation of residues in FR2 region both in interacting with antigen and shaping the CDR3 loop [27,35,36,38,39,40]. Collectively, immunogenetic and structural data support the notion that evolutionary mechanisms have actively been involved to diversify the VHH repertoire, and increase the contribution of non-CDR residues in VHH:antigen interaction compared to classical VHs, therefore compensating for the absence of VL domain [41].

In regard to the evolutionary emergence of HcAbs, several hypotheses have been proposed though it is difficult to draw solid conclusions. The most plausible explanation for the existence of HCAbs is that there is an evolutionary advantage associated with this class of antibodies. The unique paratope structure, which is smaller and with a more prolate shape than conventional mAbs, may provide better access to cryptic epitopes (e.g., catalytic clefts of enzymes or viral cavities). Indeed, the number of reported VHHs to be enzyme inhibitors or virus neutralizers well exceeds that of similar reports for conventional antibodies. Given the relatively recent discovery of VHHs in comparison to mAbs, this finding likely indicates that these domain antibodies are well-adapted to target a few sites of vulnerability by recognizing highly conserved epitopes among different strains of viruses (e.g., cross-reactive VHHs against influenza A and B strains) [20,36,41,42,43,44]. Additionally, the presence of non-canonical intradomain disulfide bridges improves the stability of the long CDR3 loops and the entire VHH domain, which makes HCAbs better equipped to tolerate harsh physiological conditions [45].

Genomic studies have shown that both classical and HCAb variable domains (VH and VHH) are encoded by IGV genes, with a distinct but intermingled set of genes for VHs and VHHs. These minigenes recombine with the same IGD and IGHJ minigenes to generate a single VDJ gene fragment for transcription for both VH and VHH. There is an identical VH and VHH gene organization between camelid species with an estimated number of 17 VHH minigenes for alpaca and 42 VHH minigenes for dromedary [46]. Homology analyses of alpaca genomic and cDNA sequence showed that there are at least three V subgroups in the *igh* locus: IGHV1 (homologous to VH families 2, 4, and 6 in the human IGHV clan II); IGHV2 (equivalent to the human IGHV clan II with VH families 1, 5, 7); and IGHV3 (equivalent to the human IGHV clan III with human VH3 family). All the identified VHH genes in alpaca are homologous to the human IGHV3 and clustered based on sequence similarity into six subsets [46]. cDNA sequence analyses in llama have shown that VHH sequences can be further divided into four subfamilies by sequence similarity, with each subfamily displaying one or more of the features characteristic for VHHs discussed earlier, such as on average longer CDR3s, extra disulfide bonds, and the existence of novel canonical structures for all three CDR loops [17,45,47]. Sequence alignments with the Kabat mammalian antibody database [14] showed that human VH3 of clan III has the highest match with the camelid VHHs in framework regions, except for the VHH hallmark residues in FR2 which are the genomic imprint of the VHH minigenes [48].

Additionally, cDNA library analysis of the llama IgG repertoire identified a set of V genes with a distinct leader signal and a high degree of homology to clan II (human VH4 family). These VH genes do not have the imprint of VHH solubilizing substitutions in FR2; however, the VDJ product can be rearranged to encode either the heavy chain of conventional antibodies or HCAbs. Therefore, these V domains could contribute to the diversity of both conventional and heavy-chain repertoire. Due to the high homology (79–89%) to human VH4, it is suggested that these antigen-specific VHs could bypass the humanization step often performed on VHHs for human applications. This potential was first demonstrated by the identification of VH4 domains with antigen-binding specificity and good solubility behavior against CD11c^+^ on dendritic cells by phage panning of an immune llama library [49].

The constant domain is encoded by three distinct genes in dromedaries. IGHG1 encodes the conventional IgG1 constant domain with the CH1 domain. Heavy-chain IgG2 and IgG3 isotypes are encoded by IGHG2 and IGHG3. These isotypes harbor point mutations at the 5′-end of the CH1-hinge intron (G to A) which abolishes the existing consensus splicing site (GT) in the IGHG1 gene, leading to the deletion of the CH1 domain in the course of mRNA splicing [27,34,36,39,46]. In llama and alpaca, additional IGHG genes including IGHG1A, IGHG1B, IGHG2A, IGHG2B, IGHG2C, and IGHG3 have also been reported, with IGHG2B and IGHG2C lacking CH1 in the alpaca genome [35,39,46,50].

## 4. Development of Camelid Single-Domain Antibodies as Therapeutics

With technological advances in the 1980s and the identification of the genes encoding Ig molecules and the molecular mechanism of Ig development in B cells [51], the idea of downsizing IgG molecules to their antigen-binding domains (Fv or VH-VL domain) was materialized by a number of research groups [52,53]. These smaller IgG fragments were useful to expand applications of antibodies (e.g., neutralizing reagents that lack effector function) and to simplify antibody gene cloning and engineering to enable library construction and phage display technologies [54]. The intriguing idea to use the VH domain as an independent single-domain antigen-binding unit was first suggested by Ward and colleagues in the 1980s [55]. In a *Nature* report, the group examined the binding properties of a standalone VH domain with its Fv counterpart (VH:VL) derived from a mouse mAb (D1.3) or from an amplified VH library made from the spleen of a mouse immunized against lysozyme. The VH domains showed specific binding to the target antigen, albeit with 10 times lower affinity which may be improved by additional engineering. However, the major shortcomings were solubility and stability of the VH domains which required further genetic manipulation. Due to these shortcomings, the idea of generating sdAbs did not captured sufficient commercial interest at that time. The discovery of HCAbs by Hamers–Casterman and colleagues in camelid species in 1989 re-ignited the idea of single-domain antibodies. Due to the physical absence of a light chain, these domain antibodies have a natural, built-in design to overcome the solubility and aggregation behaviors observed with mouse and human VH domains. It took a few years of intense investigation to demonstrate that these smaller antibody formats in camelid blood are not artifacts or disease-driven antibodies (known previously as heavy-chain immunoglobulin diseases) which were reported in humans [56,57]. Confirmatory results were eventually published in *Nature* in 1993 [11]. Antibody engineering and phage display techniques were then applied to clone the distinct camelid heavy-chain repertoire and the camelid heavy-chain antibody fragment (VHH) was established as an independent antigen-binding domain with excellent biophysico-chemical properties [18].

The first movement toward using these domain antibodies as potential therapeutics started about a decade later when Ablynx, a biopharmaceutical/VIB-funded spin-off company, was established in December 2001. The major scientific developments in the first decade included: (a) extensive studies on the VHH sequences obtained by cDNA libraries and defining VHH families [13,17,47]; (b) camelid genomic studies shedding light on the location of VHH and VH gene segments and demonstrating the germline imprint of the VHH sequences [38,40,41]; (c) the application of phage display technology to isolate VHHs from immunized libraries [18] among which a few were shown to be enzyme inhibitors [58]; (d) high-resolution crystal structures of several VHH:antigen complexes, revealing important deviations from conventional antibody:antigen interactions [23,25,59,60,61]; (e) using bacterial and yeast system to express large quantities of VHH and evaluation of various purification protocols to obtain highly pure VHH proteins [62,63]; and (f) the use of VHHs as reagents in immunoaffinity purification and immuno-perfusion [64].

The second decade of VHH development (2003–2013) started with the exponential growth of publications and patent applications mainly by VIB and other subsidiary companies in Belgium, as well as by more than 50 institutions around the globe [65]. The major hallmark of this decade was the start of preclinical and clinical studies of several nanobodies by Ablynx and others as therapeutics and imaging reagents [66,67,68], including VHHs against: (a) blood glycoprotein vWF to control platelet aggregation and clot formation [69]; (b) viral infection (RSV) [70]; (c) venom toxins [71]; (d) IL6-R for treatment of rheumatoid arthritis [72]; and (e) the use of radio-labeled nanobodies for HER2+ tumor imaging [73]. Major technological advancements were also made in the expression of VHHs in heterologous systems and in creating an array of bi- and multivalent VHHs with superior efficacy during this decade [27,74].

Since 2014, several major developments have dramatically increased research activities around the camelid VHHs. Most importantly, the first positive phase I trial results of Ablynx’s lead therapeutic domain antibody (Caplacizumab) was published and established the proof-of-concept for the first application of a nanobody in medicine. Around the same time, more VHH-based therapeutics were advanced into clinical trials and Ablynx expanded its collaborations with large biopharmaceutical companies such as Merck, Boehringer Ingelheim, Sanofi, etc., with more than 20 pre-clinical and clinical programs. Another development was the expiry of a main camel patent in 2014 in Europe and in 2017 in the US. The turning point in the history of Ablynx occurred in January 2018 when French drugmaker Sanofi announced that it had acquired the Belgium biotech company at a price of $4.8 billion. Following this and toward the end of 2018, the first nanobody drug (caplacizumab; bivalent anti-vWF nanobody for treating rare blood clotting disorders) received EMA approval (with FDA approval shortly after in February 2019) and entered the market (www.sanofi.com (accessed on 11 February 2022)) [75,76]. Caplacizumab is the eleventh antibody fragment approved by the FDA after a number of Fabs and scFvs (based on conventional mAbs) entered the clinic in the past two decades [77]. As intellectual property restrictions on the VHH composition was gradually subsiding, the interest has grown by biopharma and biotechnology companies to investigate the potential applications of VHH domains in areas of medicine (both as therapeutic and diagnostic reagents), industry and research [68,78]. For example, Taisho Pharmaceutical in Japan has used an anti-TNF VHH generated by Ablynx and developed a humanized trispecific nanobody (two anti-TNF VHHs linked to an anti-HSA VHH). The therapeutic drug, ozoralizumab, is now under review by Japan’s Pharmaceuticals and Medical Devices Agency (PMDA) [79]. As of 2021, more than 25 companies are working directly or in collaboration with Ablynx (Sanofi) on different areas of VHH development, ranging from services in isolation and building libraries to using VHHs as diagnostic moieties and therapeutic agents (Table 1). At the same time, there are 16 therapeutic nanobodies that are currently in clinical trials or that will soon enter clinical trials under the sponsorship of large pharmaceutical companies such as Sanofi (Ablynx), Novartis, Boehringer Ingelheim, AbbVie, and Merck [76] (Study: NCT03322735; clinicaltrials.gov (accessed on 18 February 2022)).

## 5. Camelid sdAbs: Pros, Cons, and Applications

There are a number of approaches to isolate camelid sdAbs against a target of interest. The starting point is to capture the heavy-chain repertoire of a camelid and build a library for screening purposes. Currently, the dominant approach is to immunize camelid species and build the library based on the repertoire of heavy-chain immunoglobulins [81]. However, reports of generating naïve libraries as a more universal source of isolating VHHs against virtually any target antigen have been published, although the affinity of the isolated antibodies is often in the (micromolar) µM range and rarely reaches the nanomolar (nM) range [82,83]. Synthetic and semi-synthetic VHH libraries are other alternative approaches; these rely on the use of a universal well-characterized VHH scaffold with randomized CDR regions [84,85,86]. Rational design of the synthetic library is necessary, especially since bacterial transformation does not capture all of the randomized variants. For example, a recent report designed a large humanized synthetic VHH cDNA library by taking into account the alpaca germline sequences for CDR1 and CDR2, structural data available in PDB for CDR3 loop structures (classified as upright, half-roll, and roll), and some key residues in FR2 region that are important in the formation of the paratope. This library was used to successfully isolate VHHs with low nM affinities against two cancer antigens using an in vitro translation screening approach [87].

One advantage of camelid VHH antibody library construction from immunized animals is that, by using a proper set of primers, the cloned immunized VHH repertoire almost mirrors the heavy-chain repertoire which has been matured in vivo. On the contrary, single-chain variable fragment (scFv) library construction requires the artificial joining of VH and VL domains by a synthetic linker which may lead to challenges with stability and proper folding, while also losing the original VH-VL pairing, requiring a larger library size to capture all combinations. Being a single domain, VHHs bypass a number of complexities related to the library construction, panning, and expression of conventional antibody fragments such as Fab and scFv as these domain antibodies do not require synthetic assembly of VH and VL domains or chain association. The VHH libraries are also genetically stable and fold efficiently when displayed on phage or other selection platforms. Additionally, their expression is less cumbersome in bacterial systems as they do not require complex folding machinery. We regularly obtain 10–100 mg/L of purified VHHs in bacterial systems, and this is considered the norm in many published works [88,89]. As mentioned before, protein expression, including VHHs, is a complex process and many intrinsic and physiological factors including the gene sequence, plasmid constructs, host expression, and folding machinery as well as expression conditions impact the yield and functionality of the recombinant protein [90]. We and others have also produced several VHHs in more labor-intensive mammalian systems with no obvious advantages in terms of yield or domain activity [91]. However, when it is required that a VHH be expressed as fusion to a human Fc for certain application such as increasing its half-life or engaging other immune cells, then mammalian expression systems are considered the best option [92]. It has also been shown that over 100 mg/L quantities of VHHs could be produced in yeast systems and this may have important biotechnological implications when large quantities of VHHs are needed to be produced at low cost [93,94]. For the purification and polishing of the final product, the customary method has so far been the use of His-tag and this has worked quite well. However, as VHHs have high homology to the human VH3 which has shown to have the capability of binding to protein A [95], we showed that about a third of camelid VHHs are naturally protein A binders and could be purified with a protein A column. We also demonstrated that a non-protein A binding VHH could be mutated in non-CDR regions to re-gain its protein A binding properties [96]. Therefore, VHHs could be produced in a His tag-free format and purified by the golden standard protein A purification system regularly practiced by the industry for the purification of mAbs.

There are a number of inherent features which make VHHs ideal for many applications, such as cancer therapy (as immunotoxin or radioisotope conjugates) and tumor imaging. The small size of VHHs gives them a unique advantage for rapid extravasation, deep tumor tissue penetration, and rapid tissue/blood clearance [76,97]. The most notable example is the development of a radiolabeled anti-HER2 VHH conjugated with 68Ga-NOTA to detect brain metastasis in breast cancer patients using PET/CT which is currently in a phase II clinical trial (NCT03331601) [98,99].

There have been a number of studies on the pharmacokinetic (PK) and pharmacodynamic (PD) profiling of VHHs, with the most comprehensive one reported on caplacizumab by Sanofi [100]. The PK and PD study of antibodies, in particular VHHs, is quite complex and requires intensive investigation and a detailed discussion is out of the scope of this review. In brief, factors which affect the PK of therapeutic antibodies are the route of injection and absorption, antibody distribution, antibody clearance, and anti-therapeutic antibody responses by the host immune system [101]. For instance, VHHs are shown to have short-half lives and rapid clearance by the kidney due to their small size and several methodologies have been applied to overcome these limitations including dimerization/multimerization, PEGylation, fusion to human serum albumin, or to a second anti-serum albumin binding VHH [102,103,104,105,106]. VHHs are expected to have minimal immunogenicity during therapeutic use in humans due to high homology (86–94%) with the human VH3 family [48,68,76,107]. However, genetic engineering approaches to humanize camelid VHHs have been developed to further reduce the immunogenicity of VHH domains [108,109,110,111].

Additional features of VHHs include their high thermostability (high refolding capacity) and relative resistance to harsh environments such as the GI tract and chemical denaturation, non-physiological pH, and high pressure. These features make them ideal reagents for oral and aerosol therapy, antibody-drug conjugates, and immobilization on microchip biosensors [76,99,112,113].

Today, VHHs have entered almost all biological research fields and there are countless reports on the isolation and characterization of the VHHs against numerous disease biomarkers specific to cancer, biochemical and hematological disorders, neuroinflammation, and infectious pathogens [68,76]. As mentioned before, VHHs, due to their small size and special surface topology with protruding CDR3, are able to recognize epitopes that are otherwise inaccessible by larger antibody fragments or mAbs and, at the same time, bypass the drawbacks related to small-molecule drugs being lack of specificity and off-target toxicity [114]. These may also include epitopes on difficult-to-access targets such as intracellular proteins [115,116], ion channels [117], and G-protein coupled receptors [118,119], and cryptic epitopes on the surface of pathogens [120].

Another potential application of VHHs is the treatment of neurodegenerative diseases. The strong target specificity of antibody-based therapeutics is a major advantage compared to small molecule drugs for the treatment of neurodegenerative diseases. However, the delivery of protein therapeutics to the brain has proven challenging, since only a small fraction of injected mAbs (0.1–0.2%) reaches the brain due to the tight blood–brain barrier (BBB) [121]. Bispecific antibodies targeting transferrin receptor have facilitated the transmigration process by shuttling binding proteins across the BBB resulting in a 5–10-fold increase in brain uptake [122]. VHH-based therapeutics with an ability to transmigrate the BBB may open a new window of opportunity for brain research and treatment. For example, by applying a proper selection/panning strategy, several VHHs were isolated from phage libraries which have the ability to transmigrate human brain endothelial cell layers using a receptor mediated process and transfer the attached therapeutic cargo into the brain in rodents [123,124,125,126,127,128]. Several VHHs isolated from naïve llama or immune libraries (e.g., FC5, FC44, anti-IGF1R5, and anti-VCAM VHHs), alone, in bispecific, liposome-mediated, or Fc-fused formats were able to improve the delivery of target peptides/cargo up to 10–30-fold when compared to the control with no VHH [129,130]. A VHH-Fc construct with a longer half-life in blood circulation led to improved delivery of the peptide although the advantage of small VHH size is compromised [121]. Extensive research to identify ideal targets and to improve antibody engineering is still required to reach the maximum delivery of antibody-cargo into the brain. The FDA has recently approved the first biologic drug to treat Alzheimer’s disease, aducanumab; a human IgG1 mAb developed by Biogen. This drug clears β-amyloid clumps in the brain, which is considered by some neuroscientists as one of the hallmarks of Alzheimer’s disease. However, a relatively high dose of the antibody (600–750 mg/month) is required to be injected intravenously with only 1.5% penetrating into the brain, reaching a peak level over five months [128,131,132], suggesting that there is room for improvement if BBB penetration could be increased.

VHHs are ideal antigen-binding domains for intracellular expression particularly due to relatively simple folding requirements in the reducing environment of the cytoplasm. For such applications, a pre-selection strategy is required to choose those VHHs with no additional cysteines and better folding capacity. These features, combined with the preferred recognition of conformational or discontinuous epitopes, make VHHs very promising tools to visualize and monitor the conformational state of proteins inside cells. VHHs could also be used to disrupt cellular pathways, and target certain disease targets inside the cells and intracellular pathogens. There are multiple approaches by which VHHs could be delivered inside cells: these include disruption of the cellular membrane by physical methods such as microinjections, electroporation the use of cell-penetrating peptides and polycationic resurfacing, DNA transfection, liposome-mediated mRNA delivery, or viral transfer of DNA into cells [133]. For instance, fluorescently labeled VHHs (chromobodies) have had a huge impact on understanding protein–protein interactions and tracking and analyzing the behavior of target proteins in vivo [134,135,136,137]. Numerous examples of VHHs as blocking reagents when expressed intracellularly have been reviewed by Soetens and colleagues [133]. Examples range from inhibiting cellular pathways to preventing the development of diseases, such as cancer and neuroinflammation, to blocking viral and bacterial multiplication cycles (e.g., HIV, HCV, HBV, RSV, *Salmonella enterica*) [89,133,138,139,140].

The innate modular and monomeric nature of VHHs makes them ideal building blocks for the generation of bi- and multispecific antibodies and for multidomain functional molecules such as those used in CAR-T and other cell therapies [68,76,78]. The small size of VHHs allows for great engineering opportunities to assemble multiple domains with unique specificities without having issues reported in other multidomain antibody fragments, such as aggregation and immunogenicity [141,142,143]. VHHs with unique specificities can be linked genetically to interact with multiple tumor targets on cancer cells to increase therapeutic efficacy to bridge multiple surface receptors or ligands to block or activate synergistic signal pathways on a cell, or to trigger contacts between cancer cells and immune cells [142,144]. The modularity of VHHs allows for the linking of multiple domains including anti-human serum albumin (HSA) moiety, to extend their half-live simultaneously, while still maintaining stability and high production yields [102].

The bispecific antibody market is growing very fast and is valued at over $20 billion USD. Blinatumomab was the first bispecific T-cell engager (BiTE), which uses two scFv fragments to target CD19 on a cancer cell and CD3 on a T cell, and was developed by Amgen and approved in 2014 [145]. Similarly, a number of therapeutic nanobodies, in bi-/multi-valent or bi-/multispecific formats have advanced into pre-clinical and clinical development by Ablynx/Sanofi and other biopharmaceutical companies thus far [68,76]. Another important application of VHHs is in the development of single or bispecific/biparatopic CAR-T cells. Conventional CAR-T cells using scFv fragments may face challenges in proper domain assembly/orientation upon display on T cells and the use of VHHs is expected to improve the expression efficiency of the recombinant T cell receptor. Indeed, the first VHH-based CAR-T (ciltacabtagene) with two anti-BCMA VHHs in its binding domain was approved by the FDA in March 2022 for the treatment of refractory multiple myeloma [146]. There is also a second clinical BCMA CAR-T candidate in clinical trial which is registered by Henan Cancer Hospital [76], (Study: NCT03322735; clinicaltrials.gov (accessed on 27 February 2022)).

Applications of VHHs as a building block to engage other types of immune cells, such as natural killer (NK) cells in bispecific formats have also been demonstrated [147,148,149]. NK cells are part of the innate immune system and have the advantage of directly engaging target cancer cells without involvement of the MHC complex, reducing toxicity related to off-target CD8+ T cell responses, and a higher proportion of NK cell infiltration within the tumor microenvironment. These features make NK cell engagers one of the most powerful and promising therapeutic reagents for treating hematological malignancies and solid tumors [77]. The feasibility of generating VHH-based BiKE molecules with anti-CD16 and anti-EGFR/HER2 arms was shown in a recent study where the BiKE molecule is able to activate and trigger degranulation of primary NK cells as measured by interferon Ɣ and CD107a expression [104]. It is expected that VHH-based BiKE and TriKE designs, as well as CAR-NK constructs, will dominate the future biotherapeutic market in years to come due to ease of engineering and production yield [77,99].

Infectious diseases and emerging pathogens are other important research areas in which VHHs have unique applications. As discussed earlier, structural studies on VHH:antigen complexes showed that VHHs tend to target more rigid, conserved, and structured epitopes. This has important implications for the development of species cross-reactive VHH therapeutics against infectious disease targets which constantly mutate to escape the host immune system. Large numbers of VHHs against bacterial and viral pathogens have been isolated and published, with a number of them in different phases of clinical trials [43,150]. During the COVID-19 pandemic, efforts were intensified to find passive immunotherapy reagents that block/neutralize SARS-CoV-2 entry. Using phage or ribosome display technologies, many VHHs with the ability to block or neutralize SARS-CoV-1 and SARS-CoV-2 have been isolated from naïve, synthetic/semi-synthetic and immune llama libraries, or from the libraries built from transgenic mice carrying camelid VHH repertoires, in a relatively short period of time [151,152,153,154]. As of July 2021, there have been 664 sdAb entries in the Coronavirus Antibody Database (www.opig.stats.ox.ac.uk (accessed on 29 February 2022)) from more than 40 research groups around the world, including 65 patent applications to protect SARS-CoV-2 sdAbs. One highlight was the work by Güttler and colleagues where 45 VHHs were isolated from an immune alpaca library with all VHHs shown to block viral infection [155]. It was shown that some VHHs, when used in tandem, could tolerate immune-escape mutations found in almost all SARS-CoV-2 lineages. A subset of these VHHs binds to either the open or closed state of the spike protein and receptor-binding domain (RBD), neutralizing the virus at picomolar (pM) concentrations with superior thermal stability (up to 95 °C). Interestingly, this group also reports some VHHs with the ability to assist the folding of the RBD domain in *E. coli*. This finding paves the way toward expressing this difficult-to-express protein in a simple and low-cost bacterial system for future vaccine production [155], and may prove to be more widely applicable to other difficult-to-express antigens.

Despite the many advantages mentioned above, there are some limitations related to further engineering of the VHHs or certain applications where VHHs are not ideally suited. The single-domain nature of VHHs, with about 110–130 amnio acids, makes the involvement of each and every residue more significant when compared to the two-domain structure of scFvs. Consequently, highly accurate engineering along with modeling studies for the purpose of, for example, humanization will be required in order to preserve the integrity of the VHH domain. Our experiences on VHH humanization have shown that only a limited number of modification is allowed, and a complete humanization often led to a drastic decrease of affinity, hampering the stability of the scaffold, and/or reduction of expression yield [156].

VHH scaffolds may not also be optimally designed to detect or capture small molecules in vitro and in vivo. The dominant convex surface topology of a VHH scaffold could not ideally accommodate molecular interactions when compared to the flat or concave topologies provided by conventional antibody fragments such as scFvs and Fabs. Although repeated immunization of llamas and alpacas has resulted in binders with low to high nanomolar affinities against several haptens and small molecules [157,158,159], it remains to be seen if these VHHs would find a marketable application. It is obvious that high stability and being resistant to harsh environmental conditions make VHHs ideally suited for many immunodiagnostic platforms for detecting small environmental pollutants; however, low nM to pM affinity antibodies will be required for such applications which seems to be challenging, though not impossible [28,78,160].

Another limitation in working with camelid species is logistics. The camelid species may not be readily accessible to every researcher and animal husbandry and immunizations per se require special facilities. For some immunization procedures such as DNA immunization, it is important to use more than one animal and analyze the consistency or heterogeneity of immune responses. Outbred llamas are not ideal for this type of assay establishment. We and others have used multiple DNA immunization procedures in llamas and it is very difficult to evaluate the efficiency of each methodology and its reproducibility. Despite these difficulties, VHHs with high affinity and specificities against difficult-to-access targets such as membrane proteins have been reported with no detailed immune response analysis. It seems that the lack or presence of a polyclonal immune response against the target of interest is not a valid indicator for library construction in llamas following DNA immunization [161,162,163,164]. To overcome this limitation, transgenic mice strains have been established, which harbor either a rearranged dromedary γ2a chain or hybrid llama/human antibody loci, and have been shown to produce functional camelid or hybrid llama-human heavy-chain antibodies [153,165,166,167]. The most notable example is the report by Xu J and colleagues [153] where transgenic mice harboring alpaca, dromedary, and Bactrian camel VHH repertoires were used to isolate potent neutralizing VHHs against SARS-CoV-2 variants which recognize conserved epitopes on RBD domain otherwise inaccessible by conventional antibodies.

## 6. Monoclonal Antibody Market and Camelid Single-Domain Antibodies: Promises and Challenges

The first therapeutic mAb, muromonab-CD3 (trade name: Orthoclone OKT3) for the prevention of kidney transplant rejection was approved by FDA in 1986 for human use. This is a little more than a decade following the invention of hybridoma technology by Kohler and Milestein [168,169,170]. According to Research & Market Report in April 2021, the compound annual growth for mAbs is close to 12% and the projected global sales of mAbs are approximately $114 billion (https://www.researchandmarkets.com/reports/5319143/monoclonal-antibodies-mabs-global-market-report (accessed on 21 February 2022)). As of December 2021, over 100 mAbs and antibody fragments have been approved by the FDA in the USA and/or by the EMA in Europe for different disease indications ranging from cancer to autoimmune and chronic inflammatory diseases to infectious diseases [3,77,171].

The lack of restrictive IP on the original hybridoma technology for production of mAbs was a major incentive to explore the great potential of monospecific biologics in various field of research and to develop effective research tools, diagnostic reagents and therapeutics. Advances in molecular biology and immunology in the 1980s provided the opportunities to clone antibody genes from hybridoma cells/immunized B lymphocytes and express them in prokaryotic and eukaryotic systems. Therefore, “second generation” of antibodies including Fabs and scFvs were made known. This, along with the introduction of phage display technology in the late 1980s, opened new windows of opportunity to isolate antigen-specific antibody fragments from phage-displayed libraries with direct access to their genetic materials [172,173,174]. These technological advances led to the introduction of the first fully human mAb adalimumab (brand name: Humira), which was approved by FDA in 2002 for the treatment of rheumatoid arthritis [54,175,176]. As of March 2022, 12 FDA-approved therapeutics including Fabs, scFvs, VHHs, and scFv-based and VHH-based CAR-T cells entered the biologic markets with many more in different phases of clinical trials [77,146,171].

A survey of the literature clearly shows that camelid VHHs are the “third-generation” of antibodies that have many biotechnological advantages over the conventional antibody fragments (Fab and scFv) and are forecasted to dominate the biological markets in the years to come. Indeed, the commercial applications of VHHs for non-medical applications appeared first in the market before its therapeutic application (caplacizumab), which was approved by EMA and FDA and entered the market in 2019 [75,177,178].

Other than their potential application as research tools, which would complement the role of mAbs, extensive efforts by many research groups around the world have been performed to isolate sdAbs for diagnostic and therapeutic applications. A countless number of excellent reviews have summarized the application of sdAbs as research reagents and medicine following the establishment of Ablynx back in 2001 [27,68,76,77,78,99,121]. The expectations for camelid VHHs two decades after their discovery are very high. Many in the research community and pharmaceutical industry believe the many advantages of camelid sdAbs will enable relatively cheap VHH-based diagnostic and therapeutic products to flow into the market and rival existing mAb and conventional Ab fragments. Surprisingly, there has been limited critical debate in the literature on why these expectations have not been met almost three decades after the discovery of VHHs. Thus far, the first therapeutic sdAb on the market has an average course of therapy costing ∼$270,000. Based on a cost analysis published recently [179], the VHH-based drug does not appear to be cost-effective in its current application, although it is considered an important breakthrough in the treatment of immune TPP therapy with a clear improved outcome for patients. In comparison, the first mouse-derived mAb was reported in 1975 and the first therapeutic mAb entered the market in 1986. Since then, 100 therapeutic mAbs and antibody fragments have been approved by FDA/EMA and other regulatory agencies and it continues to appear that conventional Abs and fragments thereof will continue to dominate the biopharmaceutical market for years to come (Table 1). Does this mean that something has been overlooked in the past two decades and needs to be revisited, or is it simply that the best niche for sdAbs must be identified? Certainly, intellectual property (IP) restrictions have hampered the ability to freely work on these domain antibodies between 2001 and 2017. It is also clear that VHHs have advantages in many applications such as bispecifics or CAR-T and one VHH-based CAR-T just hit the market in 2022 [146]. However, the original assumption presenting VHHs as cheaper alternatives to mAbs has not materialized for the first marketed VHH. The research and regulatory pathway to generate VHH-based therapeutics is the same as for mAbs and most of the associated costs in developing a biologic drug is accrued through this development period, as opposed to ongoing costs associated with manufacturing, etc. There is great hope that VHH-based therapeutic and diagnostic reagents will make significant contributions to the biologics market in the next decade or so. Several recent events (breaking the market barrier with the first VHH in 2019, the increased involvement of numerous mediums to large sized pharmaceutical companies, and the expiration of the main camel patents) may be the impetus needed to expand the market share of VHH-based therapeutics more rapidly, finally realizing some of the original promises in years to come.

## 7. Concluding Remarks

After almost three decades of the discovery of camelid heavy-chain antibodies by professor Hamers and his research team in Belgium, countless research articles and patents have been published and the commercial fruit of their discovery is gradually entering the pharmaceutical market. It is noteworthy to mention that the initial picture after the first observation of HCAbs in Belgium was quite gloomy and it took several years of extensive research to demonstrate that these antibodies were naturally-derived and not a disease by-product. However, as cDNA sequencing and genomic and structural data accumulated, we were able to piece the puzzle together and draw a beautiful picture of what nature had designed, now known as “VHHs” or “camelid sdAbs” or “nanobodies” [15,18,78]. It is not unrealistic to consider the emergence of camelid single-domain antibodies with many unique features as a revolution in the field of antibody engineering and applications. It is clear from a recent survey of the literature that an increased number of research groups and commercial entities around the globe are actively working independently or collaboratively on these domain antibodies. Similar to the classical mAbs, VHHs are being developed for various applications in the field of infectious diseases (e.g., SARS-CoV-2), cancer (e.g., bispecifics and CAR-Ts), CNS diseases (e.g., BBB drug delivery), and as intrabodies. With the approval of the first VHH-based therapeutic (caplacizumab) and an additional 16 VHH-based drugs in various stages of clinical development [74], it appears that VHHs are gradually finding their niche in the biologics market despite a number of earlier challenges and setbacks. It is predicted that within the next decade many more VHH-based therapeutic and diagnostic reagents will enter the market, hopefully with a lower price tag and of superior, or at least complementary activity, to classical mAbs and fragments thereof.

## Figures and Tables

**Figure 1 ijms-23-05009-f001:**
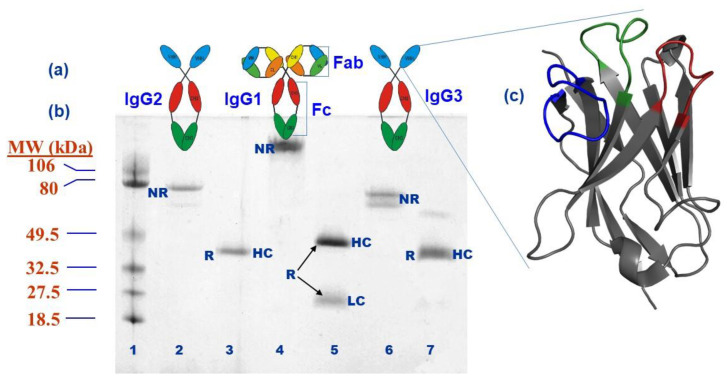
Schematic representation of camelid IgGs and llama serum IgG fractionation on protein A and protein G. (**a**) The comparative structures of each respective IgG isotype has been shown on top of lanes 2, 4, and 6. (**b**) Llama immunoglobulin serum was fractionated on protein G and A and ran on reducing and non-reducing SDS-PAGE. Lane 1: MW Marker; lane 2: IgG2 (Protein A) (non-reduced: NR); ); lane 3: IgG2 (reduced: R); lane 4: IgG1 (Protein A&G) (NR); lane 5: IgG1 (R); lane 6, IgG3 (Protein A&G)) (NR); lane 7: IgG3 (R); (**c**) the VHH folding structure of two β-sheets with five and four β-strands is shown on the right with CDR loops shown in dark green (CDR1), red (CDR2), and blue (CDR3).

**Figure 2 ijms-23-05009-f002:**
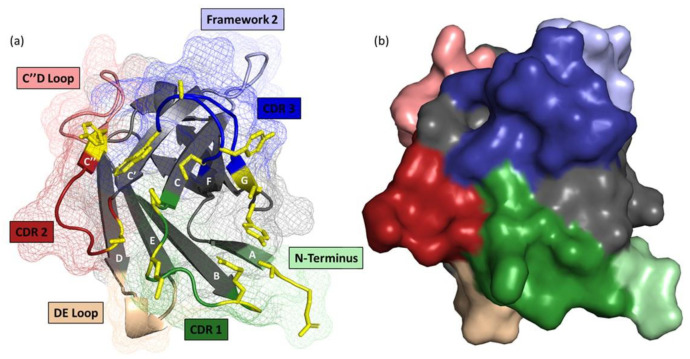
Visualization of both CDR and non-CDR contacts in nanobodies. VHH A.20 (PDB 4NBX; approximately 2.5 × 4 × 3 nm in size) is shown as a (**a**) cartoon with an overlaid mesh and (**b**) surface representation. The molecule is orientated with a view looking down on the paratope, with CDR1/2/3 colored in dark green, red, and blue, respectively. The flexible non-CDR regions are highlighted in light colors, including the N-terminus (green), framework 2 (blue), C’’D loop (red), and DE loop (orange). The contact points in CDR and non-CDR hotspots for VHH A.20 with Toxin A are highlighted as yellow sticks, and β-strands are labeled A through G. Structural assignments are based on Zavrtanik et al. (2018).

**Table 1 ijms-23-05009-t001:** Antibody fragments approved by the FDA and/or EMA for various disease indications as of March 2022 ^#^.

Name/Brand Name (FDA/EMA Approval Year)	Sponsor	Antibody Format/Production or Expression Host	Target	Indication(s)	Cost/Injection/Month or Year Treatment
Digibind (1986)	GlaxoSmithKline	Ovine Fab/ Mammalian *	Digoxin	Digoxin overdose	$4324/I
Abciximab/ReoPro (1994)	Centocor	Chimeric Fab/ Mammalian *	GPIIb/IIIa	Cardiovascular	$19,389/Y
CroFab (2000)	Protherics	Ovine Fab/ Mammalian *	Crotalidae venom	Pit viper envenomation	$3198/Y
DigiFab (2001)	Protherics	Ovine Fab/Mammalian *	Digoxin	Digoxin overdose	$4324/I
Ranibizumab/ Lucentis (2006)	Genentech	Humanized Fab/ *E. coli*	VEGEF	Neovascular (wet) age-related macular degeneration	$6000/3M
Certolizumab pegol/ Cimzia (2008)	UCB	PEGylated humanized Fab’/ *E. coli*	TNF-α	Crohn’s disease, rheumatoid arthritis	$17,277/Y
Anascorp (2011)	Rare Disease Therapeutics	Equine F(ab’)_2_/ Mammalian*	Centruroides venom	Arizona bark scorpion envenomation	$4489/I
Blinatumomab (2014)	Amgen	BiTE(scFv-L-scFv)/ *E. coli*	CD19-CD3	Acute lymphoblastic Leukemia (ALL)	$178,000/Y
Anavip (2015)	Rare Disease Therapeutics	Equine F(ab’)_2/_ Mammalian *	Crotalidae venom	Pit viper envenomation	$1220/I
Idarucizumab/ Praxbind (2015)	Boehringer-Ingelheim	Humanized Fab/ CHO	Dabigatran	Anticoagulation	$3662/I
Moxetumomab pasudotox/Lumoxiti (2018)	AstraZeneca	scFv-Immunotoxin/ *E. coli*	CD22	Hairy cell leukemia	$39,906/6M
Brolucizumab/ Beovu (2019)	Novartis	Humanized scFv/ *E. coli*	VEGF	Neovascular (wet) age-related macular degeneration (AMD)	$8508/Y
Caplacizumab/ Cablivi (2019)	Sanofi (Ablynx)	Humanized VHH/ *E. coli*	vWF	Acquired thrombotic thrombocytopenic purpura (aTTP)	$270,000/Y
Ciltacabtagene/ Cilta-cel (2022)	Janssen, Legend Biotech Corp	CAR-T-(VHH)2	BCMA	Relapsed or refractory multiple myeloma	$206,000– 265,000

* These antibody products are not recombinant and were generated in different mammalian hosts by immunization and produced by enzymatic digestion and purified by affinity chromatography. ^#^ Data extracted and classified from: antibodysociety.org (accessed on 22 February 2022), 2020; ICER.org (accessed on 22 February 2022), 2021 [77,80]. Prices are in $USD and not adjusted for inflation. Additionally, the use of antibody fragments as reagents for detoxification, in cancer therapy, and for vWF for clotting are all very different and must be taken into account.

## Data Availability

Not applicable.

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
