# Peer review of "Camelid Single-Domain Antibodies: Promises and Challenges as Lifesaving Treatments†"

_ijms, 2022, doi:10.3390/ijms23095009_

Round 1

Reviewer 1 Report

This manuscript is a well written and informative review of the state of VHH technologies and challenges today. The author provides great background on the discovery of VHHs, their genetic origins, and their unique structural aspects. That is then followed by a discussion of the many ways that VHHs have been (or may yet be) useful as therapeutics, in imaging, or in biomedical research. I recommend publication of the manuscript.

I only found a few minor mistakes to be corrected.

Line 45, add a period after the word “serum.”

Line 103, remove “This section”

Line 167, add a comma and a space between V37F/Y and G44E

Line 255, change the word “presence” to “absence”

Line 538-539, make sure the hyperlink (if in the final paper) includes the “monoclonal-antibodies” part

Author Response

All the points were considered and incorporated and corrections were made. Please see the revised version.

Thanks for the comments and insights.

Reviewer 2 Report

This review entitled "Camelid single-domain antibodies: Promises and challenges as life saving treatments" is a very interesting piece of work.

It is a comprehensive review of the development of camelid nanobodies since their discovery in 1993. This review covers many aspects ranging from the structural features of nanobodies to their multiple applications, including their genetic origins, their advantages and disadvantages as compared to conventional antibodies and their active fragments. It is particularly well documented and I'm sure that any researcher interested in these molecules will find a real interest in reading this review.

I just have a limited number of points that could help to slightly improve the manuscript

1- Nothing is said about the pharmaceutical development of such molecules. Are there some specific advantages or drawbacks as compared to conventional antibodies in terms of host cells for production (mammalian, yeast and bacteria), yield of production and purification (recombinant VHH are not expected to bind Protein A, Protein L) and polishing.

2- PK/PD could be discussed in more detail, especially in case these molecules are selected because of their monovalence.

3- The author correctly states that the first active single-domain antibody surprisingly discovered by Sally Ward et al. (1989) paved the way to new applications. However, nothing is said about natural single domain shark VNAR antibodies. These molecules are quite similar to VHH. There existence should be indicated as well as the conditions of their discovery and their advantages/disadvantages as compared to VHH.

Other points

Figure 1: some discrepancies between the legend and the text (lines 46, 58).

lines 103-108: This part is unclear.

line 314: word missing

line 409 "VHHs are also ideally suited for intracellular expression due to their small size, high solubility and stability, and ability to fold in the reducing environment of cytoplasm." I'm willing to believe it but what about disulfide bridges formation including non-canonical bridges between CDR1 and CDR3 ?

Author Response

(The authors gave the same response as above.)

Reviewer 3 Report

Although the work is well organized and scientifically sound, the overall content of the manuscript is very similar to the author’s previous work and entire sentences are copied in several places without citing and reciting references (Arbabi-Ghahroudi, 2017 published in Front. Immunol.). For reference, Please find below a "concluding remarks" section from the submitted version showing identical text (text shown in bold) to the previously published version (Arbabi-Ghahroudi, 2017 published in Front. Immunol.). Under such circumstances, the submitted manuscript cannot be considered sufficiently unique. In my opinion, the author should write a mini-review that includes recent developments in Camelid Single-Domain Antibodies that have not been covered in the author's previous work.

Concluding remarks from the submitted version showing text overlap (text shown in bold) to the previously published version:

"Close to three decades have now passed since the first observation of camelid HCAbs by Hamers and colleagues. This finding was a significant milestone in the field of antibody engineering and opened many new opportunities and applications. It was also instrumental in reviving the concept of sdAbs, which had been originally suggested by Ward et al. in 1989. The initial picture after the first observation of HCAbs in Belgium was quite gloomy as it took several years of intensive efforts to demonstrate that these antibodies are naturally-derived and not a disease by-product. However, as cDNA sequencing and genomic and structural data accumulated, we were able to piece the puzzle together and draw a beautiful picture of what nature designed, now known to all of us as “VHHs” or “camelid sdAbs” or “nanobodies”. The unique and extraordinary features of HCAbs and their paratopes (VHHs) have attracted many researchers and commercial entities to the field of antibody engineering. VHHs are now closer than ever to approval as pharmaceutical drugs to fight a wide range of diseases including cancer, inflammation, haematology and respiratory diseases, with an additional 16 VHH-based drugs in various stages of clinical development. VHHs have also been shown to be effective as anti-infectious reagents, particularly as robust reagents in passive immunotherapy, as well as for diagnostic and imaging applications. With the approval of the first VHH-based therapeutic (Caplacizumab) by EMA and FDA, it appears that VHHs are gradually finding their niche in the biologics market despite a number of earlier challenges and setbacks. It is predicted that in the next decade or so, many more VHH-based therapeutic and diagnostic reagents will enter the market, hopefully with a lower price tag, which will be either superior or, at minimum, complementary to classical mAbs and fragments thereof."

Author Response

(The authors gave the same response as above.)

Round 2

Reviewer 3 Report

Thank you for your responses.

The authors have responded satisfactorily to all the comments and made necessary changes to the manuscript.